# Laparoscopic Treatment of Bulky Nodes in Primary and Recurrent Ovarian Cancer: Surgical Technique and Outcomes from Two Specialized Italian Centers

**DOI:** 10.3390/cancers16091631

**Published:** 2024-04-24

**Authors:** Alberto Daniele, Roberta Rosso, Marcello Ceccaroni, Giovanni Roviglione, Gianmarco D’Ancona, Elisa Peano, Valentino Clignon, Valerio Calandra, Andrea Puppo

**Affiliations:** 1Department of Gynecology and Obstetrics, Azienda Sanitaria Ospedaliera Santa Croce e Carle, 12100 Cuneo, Italy; daniele.al@ospedale.cuneo.it (A.D.); peano.el@ospedale.cuneo.it (E.P.); clignon.v@ospedale.cuneo.it (V.C.); calandra.v@ospedale.cuneo.it (V.C.); puppo.a@ospedale.cuneo.it (A.P.); 2Department of Obstetrics and Gynecology, IRCCS “Sacro Cuore-Don Calabria” Hospital, 37024 Negrar di Valpolicella, Italy; marcello.ceccaroni@sacrocuore.it (M.C.); giovanni.roviglione@sacrocuore.it (G.R.); gianmarcodancona@libero.it (G.D.)

**Keywords:** ovarian cancer, minimally invasive surgery, laparoscopy, nodal metastasis, recurrence

## Abstract

**Simple Summary:**

Epithelial ovarian cancer is typically diagnosed in the advanced stage and, despite cytoreductive surgery and medical treatments, many patients experience relapses, such as peritoneal carcinomatosis, parenchymal progression, nodal metastases, multisite disease or isolated lymph nodal recurrence. The aim of the study was to describe the principles of minimally invasive surgery in advanced and recurrent ovarian cancer with nodal involvement and analyze clinical outcomes in this setting. In our series of 21 patients undergoing laparoscopic surgery for advanced or recurrent disease with nodal metastasis, the minimally invasive approach appeared to be a safe and promising technique, with low complication rates and favorable clinical outcomes.

**Abstract:**

(1) Background: Minimally invasive surgery (MIS) represents a feasible approach in early-stage ovarian cancer, while this question is still unsolved for advanced and recurrent disease. (2) Methods: In this retrospective, multicenter study, we present a series of 21 patients who underwent MIS for primitive or recurrent epithelial ovarian cancer (EOC) with bulky nodal metastasis and discuss surgical technique and outcomes in relation to the current literature. (3) Results: Complete cytoreduction at primary debulking surgery was obtained in 86% of cases. No complication occurred in our patients intraoperatively and only 11.1% of our patients experienced grade 2 and 3 postoperative complications. Notably, all the patients with isolated lymph nodal recurrence (ILNR) were successfully treated with a minimally invasive approach with no intra- or postoperative complications. (4) Conclusions: The results of our study are consistent with those reported in the literature, demonstrating that MIS may represent a safe approach in advanced and recurrent EOC with nodal metastasis if performed on selected patients by expert surgeons with an adequate setting and appropriate technique.

## 1. Introduction

Epithelial ovarian cancer (EOC) is the eighth leading cause of cancer-related deaths in women worldwide and the first cause of death from gynecological malignancies [1]. Patients with EOC typically lack disease-specific symptoms and, in about 70% of cases, they are diagnosed in the advanced stage, significantly elevating the risk of metastasis and early death; for this reason, EOC is often referred to as a “silent killer” [2]. Although radical primary cytoreductive surgery (PCS), platinum-based chemotherapy and recently introduced targeted therapies, about 25–75% of patients eventually relapse [3,4].

At the time of diagnosis, most patients with advanced disease present with peritoneal carcinomatosis, but, in a subgroup of patients, the advanced stage is determined by nodal involvement. Regarding recurrent disease, different patterns of relapse have been described: peritoneal carcinomatosis, parenchymal progression, nodal metastases or multisite disease. Isolated lymph node recurrence (ILNR) represents a specific condition and is more common in BRCA-mutated patients [5,6]. Because the outcomes of relapsed patients are poor with salvage chemotherapy, secondary cytoreduction surgery (SCS) represents a valid option to consider [7,8].

In the last few years, several studies have demonstrated that minimally invasive surgery (MIS) represents a feasible approach in the early stages of ovarian cancer [9,10,11,12,13], while debate is still open for advanced and recurrent disease [14,15,16,17,18].

The present study reports a retrospective series of patients with advanced or recurrent EOC and bulky nodes treated with a minimally invasive approach. We present and discuss the surgical technique and outcomes in relation to the current literature.

## 2. Materials and Methods

### 2.1. Study Design

This was a retrospective, multicenter study that included patients diagnosed with primitive or recurrent EOC with bulky nodal metastasis treated with MIS between January 2020 and September 2023 in two Italian centers: “Santa Croce e Carle Hospital” in Cuneo, an ESGO-accredited center for the treatment of ovarian cancer, and “Sacro Cuore Don Calabria Hospital” in Negrar.

We describe our laparoscopic technique and surgical and oncologic outcomes considering the current literature on MIS in ovarian cancer.

Multiple data were collected: age, BRCA status, primary or recurrent disease, radiological data, FIGO clinical stage, type of surgery (primary debulking surgery (PDS) or interval debulking surgery (IDS)), site of metastatic nodes, number of removed nodes, maximum diameter of the removed nodes, length of the surgical procedure, blood loss, need for transfusion, intraoperative or postoperative complications, length of hospital stay and time to adjuvant therapies. Our definition of “bulky nodes” included size ≥10 mm, radiological abnormal appearance and/or positivity at PET scan.

In all cases of primary surgery, a preoperative CT scan was performed to clearly understand the extension and localization of the disease, while patients with recurrent disease also received a PET scan. Subsequently, each case was discussed by the tumor board, where surgeons, oncologists, radiologists, radiotherapists and pathologists were present. In our experience, a diagnostic laparoscopy was first performed to accurately examine the abdomen and evaluate the presence and extension of peritoneal carcinomatosis. In cases of resectable disease, the presence of large pelvic masses, extended carcinomatosis, the need for multiple bowel resections or anesthesiologic contraindication to laparoscopy, an open approach was preferred, whereas a previous laparotomy or the presence of diaphragmatic carcinomatosis did not represent contraindications to MIS. In recurrent disease, the minimally invasive approach was preferred in cases of single nodal recurrence or oligometastatic disease that was considered laparoscopically resectable based on preoperative imaging, tumor board assessment and intraoperative findings. The extension of the disease was intraoperatively evaluated and reported according to the Fagotti score. In all cases, peritoneal cytology was performed, but the analysis was not conducted intraoperatively, and it did not change our approach.

There are no external funding or conflicting interests to declare.

### 2.2. Statistical Analysis

The sample was described regarding its clinical and demographic characteristics using descriptive statistics techniques. Quantitative variables were summarized with median, standard deviation, min and max. Qualitative variables were described with absolute frequency tables. All the descriptions were summarized using descriptive statistics. Statistical analysis was performed using SPSS v.26.0 software (IBM Corp., Armonk, NY, USA).

### 2.3. Surgical Technique

Every laparoscopic procedure should start with a thorough inspection of the abdomino-pelvic peritoneum with the exclusion of ascites and/or carcinomatosis. In the case of pelvic nodal involvement, a lateral approach to the retroperitoneum is indicated by opening the lateral paravescical space, medially tractioning the umbilical artery and dissecting in a cranio–caudad direction until the parietal pelvic fascia covers the levator ani muscle. Subsequently, the ilio-lumbar space is developed between the psoas muscle and the external iliac vessels, which are medialized, preserving laterally the genitofemoral nerve and its genital and femoral branches. This maneuver allows safe access to the proximal portion of the obturator nerve, lumbo-sacral trunk, sacral roots S1–S4 and sacral plexus, also allowing for the hemostasis of small venous branches merging with the psoas muscle. After exposing the surgical field and identifying the anatomical landmarks, the bulky pelvic lymph nodes are progressively dissected from the adjacent vascular and neural structures by traction and countertraction along the visceral pelvic fascia covering the metastatic nodes and the vascular adventitia or perinevrium. Care must be taken to avoid injury to retropubic veins and venous anastomosis between the obturator and external iliac veins, also called “corona mortis”. To decrease postoperative lymphorrhea, meticulous coagulation is carried out and hemostatic clips are placed to interrupt major lymphatic trunks. In the case of para-aortic bulky nodes, access to the para-aortic retroperitoneum is needed to identify the exact location of the disease.

Para-caval, pre-caval and inter-cavo-aortic bulky nodes are approached by opening the retroperitoneum at the right promontorium and skeletonizing the right infundibulo-pelvic (IP) ligament (or its stump), which is gently pulled ventrally and dissected by the Gerota’s fascia covering the abdominal ureter until the origin of the IP ligament is identified [19]. The duodenum is then bluntly pushed ventrally and cranially in order to better expose the ventral face of the vena cava and aorta, thus allowing for the progressive exposure of the left renal vein. In the case of para-aortic bulky nodes, the procedure can go ahead by a further retro-peritoneal dissection exposing the lateral aspects of the aorta or (in the case of isolated left bulky nodes) it may start from the left side by opening the retroperitoneum between the IP ligament (or its stump) and the ureter. Then, the dissection is carried out up to the entrance of the IP ligament into the left renal vein. In this setting, the so-called “mesocolic window”, a space obtained by dissection of the mesosigmoid between the orthosympathetic superior hypogastric plexus (SHP) and the course of the inferior mesenteric vessels is dissected in order to further identify the course of the left ureter and preserve the main vascularization of the rectosigmoid [20]. The SHP is then pulled ventrally and lifted together with the mesosigmoid, thus preserving the nerve bundles.

The steps of the surgical procedure are shown in Figure 1 and Figure 2.

## 3. Results

Data about 21 patients who underwent surgery for primitive or recurrent EOC presenting with nodal metastasis were collected. Sixteen patients had a new diagnosis of ovarian cancer and underwent primary surgery and five patients had recurrent ovarian cancer and were submitted to SCS. In both groups, patients had bulky nodes that were removed during the planned surgery. The procedures were all completed in MIS, without switching to open surgery.

Among patients who underwent primary surgery, nine (56.3%) had PDS, while seven (43.7%) had IDS. The rate of complete cytoreduction was 100% in the case of PDS and 71% in the case of IDS. All patients who underwent SCS had ILNR and all of them had complete cytoreduction.

In our series of patients, the median age at diagnosis was 62.5 years (range 43–80) and BRCA mutations were present in 23.8% of cases, all being in the BRCA1 gene. Seventeen patients (81.0%) had high-grade serous carcinoma, two patients (9.5%) had high-grade endometrioid carcinoma and two patients (9.5%) had clear cell ovarian carcinoma.

Data about patients and tumor characteristics are reported in Table 1.

The removed lymph nodes were anatomically distributed as follows: nineteen patients with pelvic metastasis and two patients with both pelvic and para-aortic adenopathies. No patient had isolated para-aortic metastatic nodes. The median number of excised lymph nodes in patients with primary disease was six (range 1–36). Metastases were histologically confirmed in at least one of the excised nodes and, in three cases, metastases were identified in all the removed lymph nodes. The median diameter of the removed nodes was 28 mm (range 10–60 mm).

The median length of surgery was 180 min (range 85–250 min) and the median blood loss was 100 mL (range 50–150 mL). The median length of hospital stay was 5 days (range 2–12). Two patients experienced postoperative complications (grade 2 and grade 3 according to Clavien–Dindo classification): one suffered from intestinal obstruction treated with ileostomy while the other was diagnosed with pelvic fluid collection and hyperpyrexia, which was managed with antibiotic therapy. These two patients were the ones with the longer period of hospital stay, 12 and 10 days, respectively. No patient needed to be admitted to the intensive care unit (ICU) after surgery.

All patients received chemotherapy after surgery, with a median interval time between surgery and the beginning of chemotherapy of 27 days (range 13–50). All patients received platinum-based regimens in first-line or neoadjuvant therapy. Patients with germ-line BRCA mutation or HR deficiency received PARP inhibitors.

The median follow-up was 25 months (range 7–47). Among the sixteen patients who underwent primary surgery, thirteen of them (81.2%) had a recurrence and three of them were alive without disease. Among those with relapse, three (23.1%) had ILNR and ten (76.9%) had multisite recurrence. Six of the thirteen patients underwent surgical management, six patients only received chemotherapy and one received radiotherapy. Among the five patients who underwent SCS, three patients were alive with disease and two patients were alive without disease.

At the time of analysis, all patients were alive; the 2-year overall survival (OS) was 100%.

## 4. Discussion

Laparoscopic surgery has been introduced in many institutions as an alternative to open surgery for primary or secondary cytoreductive surgery, but published data are limited [2,18,19,20]. The National Comprehensive Cancer Network (NCCN) guidelines suggest that MIS can be used for selected patients for interval debulking procedures [21]. Laparoscopic treatment of ovarian cancer can be an ideal approach in highly selected patients with localized primary or recurrent disease, especially for single-site, lymph-nodal pelvic or para-aortic involvement [16].

Recent studies have confirmed that the minimally invasive approach may be a valid option for patients with advanced-stage and recurrent disease if performed by expert surgeons on selected patients [22,23,24].

Several studies showed that the IDS represents a privileged setting for the MIS, with good recurrence and survival rates. Melamed et al. found no difference in 3-year survival rates between the minimally invasive approach and open surgery cohorts, even after correcting for patient comorbidities and cancer substage [25]. However, potential confounders may be present in observational studies, such as the fact that patients selected for laparoscopic interval cytoreduction might have shown a better response to chemotherapy. For this reason, the results of the randomized Laparoscopic Cytoreduction after Neoadjuvant Chemotherapy (LANCE) trial are expected. This trial enrolled 100 patients (51 randomized to open surgery and 49 randomized to minimally invasive surgery), and preliminary data showed a similar rate of complete macroscopic resection (83% versus 87.5%; *p* = 0.6) and intraoperative complications (6.4% versus 6.3%) [26].

Gallotta et al. published a multicentric, retrospective study that demonstrated the feasibility and safety of the laparoscopic approach in 69 preoperatively presumed early-stage patients who accidentally revealed localized carcinomatosis or lymph node involvement at laparoscopic evaluation or postoperative pathological examination. Nehzat et al. reported a prospective series of 17 patients who underwent laparoscopic primary or interval debulking surgery, with 88.2% optimal cytoreduction and a median time to recurrence of 31.7 months [27]. Furthermore, Fanning et al. showed that 23 patients (92%) were successfully cytoreduced laparoscopically and showed a median overall survival of 3.5 years [28]. Moreover, a recent systematic review and meta-analysis of retrospective and prospective data by Knisely et al. indicated that laparoscopic surgery is a safe and feasible procedure in patients with advanced or recurrent EOC, finding no association between the surgical approach and overall or progression-free survival [29].

Approximately 75% of EOC patients experience a recurrence within 2 years after diagnosis [30]. Disease recurrences typically present as diffuse peritoneal carcinomatosis and parenchymal metastases or lymph node involvement. Isolated lymph node recurrence represents a specific condition accounting for about 1.1–4.2% of cases, and it is typically associated with better prognoses. The para-aortic region was found to be the most commonly involved site (43%), followed by the pelvic (33%) and combined pelvic/para-aortic (14%) regions [31,32,33]. Recurrent ovarian cancer is typically treated with systemic chemotherapy and the role of surgery has long been debated due to conflicting data. More recently, the DESKTOP-III study demonstrated that secondary cytoreductive surgery followed by chemotherapy led to an improved OS compared with chemotherapy alone [34]. Particularly, several studies showed better outcomes in patients with ILNR undergoing secondary cytoreductive surgery in comparison with patients who underwent other treatments, such as salvage chemotherapy or the irradiation of bulky nodes. For example, Ferrero et al. reported the feasibility of SCS for ILNR, achieving complete cytoreduction in 72 out of 73 cases, without significant morbidity, and a 5-year post-recurrence survival of 64% [5,35]. The quality of evidence in ovarian cancer recurrence treatment by MIS is low, being based mainly on case reports and retrospective studies with small sample sizes. Nevertheless, three studies comparing minimally invasive surgery versus the laparotomy approach showed a rate of optimal cytoreduction of 70–98% and no statistically significant differences in disease-free and overall survival rates [19,36,37].

No current guideline specifically defines the characteristics of patients who may be eligible for MIS for advanced and recurrent EOC and no predictors of its feasibility are currently available, but most case series have included patients with single-site and easily accessible disease [20,22,37]. The selection of patients is a fundamental point to successfully perform minimally invasive surgery. Conte et al. retrospectively assessed the feasibility and efficacy of this approach in patients with recurrent EOC who underwent secondary cytoreduction by laparotomy versus MIS, indicating as predictive factors for MIS the neoadjuvant chemotherapy at first diagnosis, the site of recurrence and the number of lesions. Moreover, the complete macroscopic resection rate was similar in both groups, while postoperative complications were significantly higher in the laparotomy group. The study included a highly selected group of patients who underwent stringent evaluation with preoperative PET/CT and diagnostic laparoscopy and who had the highest chance of achieving complete cytoreduction [13].

Regarding ILNR, which represents a relatively rare subtype of EOC recurrence, no standard of care is reported in the literature, and limited studies are available about the feasibility and effectiveness of the minimally invasive approach. Patients with ILNR seem to be perfect candidates for SCS as nodal recurrence was limited to one nodal region in 80–96% of cases, with a median size of 2.5 cm and a median number of two involved nodes [33]. Gallotta et al. retrospectively collected 40 patients with ILNR from different gynecological malignancies (ovarian, cervical and endometrial cancer), showing that MIS was a valid approach in very select patients [38]. Hong et al. reached the same results by analyzing six patients with ILNR (four ovarian, one cervical and one peritoneal cancer) [39]. Sanna et al. reported a case series of MIS nodal relapse asportation up to 8 cm, demonstrating that MIS for ILNR in gynecological malignancies may be a feasible, safe and effective option in terms of oncological outcomes, even for large tumors [40]. Recently, Certelli et al. published a review specifically about the role of SCS in the treatment of recurrent ovarian cancer, showing that a minimally invasive approach plays a crucial role in this setting in terms of survival rates and quality of life [41].

The laparoscopic excision of metastatic lymph nodes, especially in the para-aortic and para-caval regions, is a challenging procedure as it requires good surgical ability to remove the disease without damaging the surrounding structures, such as the aorta, inferior vena cava, renal vein, inferior mesenteric artery and ureter. The key to avoiding these complications is to provide optimal exposure of the area by gently and gradually dissecting and pushing away the mentioned structures, keeping in mind that they may be attached to the tumor and that aberrant vessels may be present [33,42].

It should be noted that tumor biology plays an increasingly important role in the surgical management of the disease. It is well known that different subtypes of EOC show different chemosensitivities. While high-grade histology is characterized by the response to platinum-based treatment in up to 80% of cases, other subtypes, such as low-grade serous ovarian cancer and clear cell or mucinous carcinoma, show a lower response rate to chemotherapy (23%, 63% and 26%, respectively). It is precisely in these histological subtypes that SCS may play a crucial role in achieving better survival outcomes [43].

The present study reports our MIS technique in patients with advanced or recurrent disease with metastatic adenopathies. In our series, complete cytoreduction at primary debulking surgery was obtained in 86% of cases, in accordance with the literature [26,29,44]. No complication occurred in our patients intraoperatively and only 11.1% of our patients experienced grade 2 and 3 postoperative complications. These rates are similar and even lower than those reported in other studies [5,23,35]. Notably, all the patients with ILNR were successfully treated with a minimally invasive approach. No intra- or postoperative complications were registered in these patients. The results of our study are consistent with those reported in the literature, demonstrating that laparoscopy may play a strategic role not only in early-stage EOC but also in advanced and recurrent disease. This method may represent an appropriate approach in a specific subset of patients with low-burden disease limited to the pelvis and lymph nodes. In both settings of primary and recurrent disease, our study confirmed the most important benefits of the minimally invasive approach, particularly the decreased complications rate, low blood loss and short hospital stay, without showing increased recurrence rates. While the short median follow-up time (25 months) limited direct comparison with the literature data, it is noteworthy that all patients in our population were alive. This suggests positive survival outcomes, though long-term observations are essential for comprehensive analysis. It is important to highlight that the presented data were derived from highly specialized oncological centers and the cited surgical procedures were performed by expert surgeons with extensive laparoscopic experience and deep knowledge of the specific anatomy of the retroperitoneum.

However, our study has some limitations, such as its retrospective nature, the small sample size and the relatively short median follow-up. Our patients were not randomized to laparoscopic or laparotomic cytoreduction but rather selected for the procedure by expert surgeons, which may account for the high rate of optimal debulking.

## 5. Conclusions

In conclusion, our study suggests that our MIS technique represents a safe approach for advanced primary or recurrent ovarian cancer with lymph node involvement if performed on selected patients by expert surgeons with an adequate setting and appropriate technique.

## Figures and Tables

**Figure 1 cancers-16-01631-f001:**
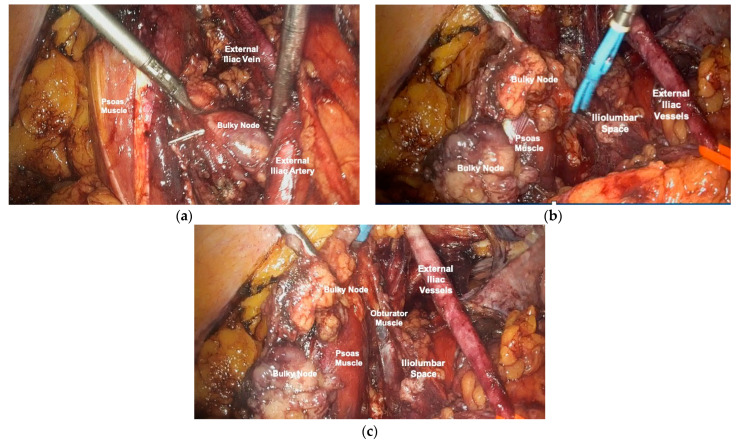
Left pelvic external iliac bulky node. (**a**) Exposure of pelvic retroperitoneum and identification of anatomical landmarks and isolation of the pelvic bulky node. (**b**) Dissection of the iliolumbar space and skeletonization of the external iliac vessels. (**c**) Removal of pelvic node with total exposure of the iliolumbar space and obturator muscle.

**Figure 2 cancers-16-01631-f002:**
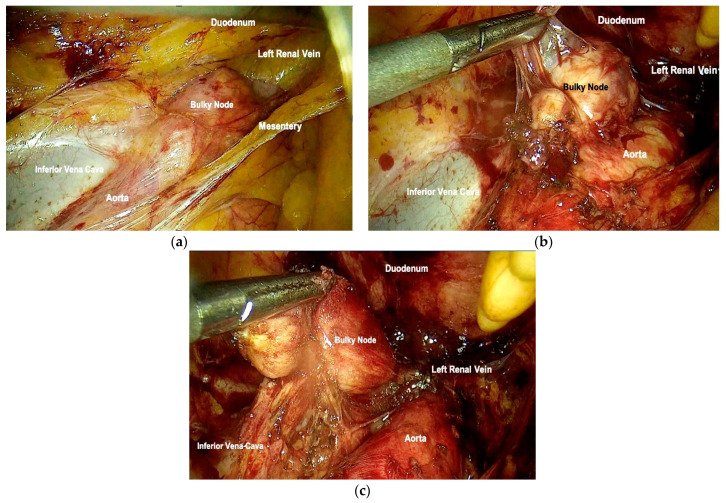
Para-aortic bulky node. (**a**) Para-aortic retroperitoneum exposed with identification of anatomical landmarks and bulky para-aortic node. (**b**) Exposure and isolation of the para-aortic node. (**c**) Removal of the lymph node.

**Table 1 cancers-16-01631-t001:** Characteristics of patients and tumors. PDS = primary debulking surgery; IDS = interval debulking surgery; SCS = secondary cytoreductive surgery; wt = wild type; AWD = alive with disease; NED = alive without evidence of disease.

Patient	Age at Surgery	Histotype	FIGO Stage at Diagnosis (FIGO 2021)	BRCA Status	PDS, IDS or SCS	Site of Adenopathy	Maximum Nodal Diameter (mm)	Patient Status
1	43	High-grade serous	IIIC	wt	PDS	Pelvic	60	AWD
2	60	High-grade endometrioid	IIIA1	wt	PDS	Pelvic and para-aortic	30	AWD
3	50	High-grade serous	IIIC	mBRCA1	SCS	Pelvic	15	NED
4	64	High-grade serous	IIIA1	wt	PDS	Pelvic and para-aortic	35	NED
5	55	Clear Cell	IIIC	mBRCA1	PDS	Pelvic	35	NED
6	52	High-grade serous	IIIA2	wt	SCS	Pelvic	50	AWD
7	65	High-grade serous	IIIA2	wt	PDS	Pelvic	35	AWD
8	67	High-grade serous	IIIA2	wt	SCS	Pelvic	30	AWD
9	57	High-grade serous	IIIA	wt	PDS	Pelvic	30	AWD
10	59	High-grade serous	IIIA	wt	SCS	Pelvic	30	AWD
11	56	High-grade serous	IIIC	mBRCA1	IDS	Pelvic	10	NED
12	60	High-grade serous	IVB	wt	IDS	Pelvic	30	AWD
13	46	High-grade serous	IIIC	wt	PDS	Pelvic	10	AWD
14	80	High-grade serous	IIIB	wt	IDS	Pelvic	10	AWD
15	46	High-grade endometrioid	IIIC	wt	PDS	Pelvic	30	AWD
16	79	High-grade serous	IIIC	wt	IDS	Pelvic	30	AWD
17	74	High-grade serous	IVB	mBRCA1	IDS	Pelvic	35	AWD
18	57	High-grade serous	IIIC	wt	PDS	Pelvic	20	AWD
19	59	High-grade serous	IIIC	wt	SCS	Pelvic	15	NED
20	59	Clear Cell	IIIC	wt	IDS	Pelvic	10	AWD
21	51	High-grade serous	IVB	wt	IDS	Pelvic	30	AWD

## Data Availability

The raw data supporting the conclusions of this article will be made available by the authors upon request.

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
