# Peer review of "Laparoscopic Treatment of Bulky Nodes in Primary and Recurrent Ovarian Cancer: Surgical Technique and Outcomes from Two Specialized Italian Centers"

_cancers, 2024, doi:10.3390/cancers16091631_

Round 1

Reviewer 1 Report

Comments and Suggestions for Authors

The reviewer is fundamentally concerned about the presented study conducted in the field of advanced ovarian cancers. Since 2019 in a NEJM (Harter et al. 2019 Feb 28;380(9):822-832.) fully published randomized trial on the importance on lymph node removal in advanced ovarian cancer clearly showed that there is no impact on patients’ course of the disease whether the nodes were removed or not.

So why did the authors conduct this study.

In a different scenario LNE removals is essential, just especially in diseases initially confined to the ovaries, addressing the role of LNE resection as a stating procedure to identify advanced cases defined by LNE involvement.

The authors also do not give an idea of how the resection of advanced peritoneal involvement was done, nor do the give an index of peritoneal spread (Sugarbaker PCI).

Furthermore, the authors already published data of the same set of patients as it appears to the reviewer.

Surgical technique for laparoscopic removal of bulky para-aortic nodes without repositioning surgical field during laparoscopic debulking for advanced ovarian cancer.

Puppo A, Olearo E, Ceccaroni M.Facts Views Vis Obgyn. 2022 Jun;14(2):189-191. doi: 10.52054/FVVO.14.2.029.

"Things Have Changed"-Laparoscopic Cytoreduction for Advanced and Recurrent Ovarian Cancer: The Experience of a Referral Center on 108 Patients.

Ceccaroni M, Roviglione G, Bruni F, Dababou S, Venier M, Zorzi C, Salgarello M, Ruffo G, Alongi F, Gori S, Driul L, Uccella S, Barra F.Cancers (Basel). 2023 Dec 6;15(24):5726. doi: 10.3390/cancers15245726.

Comments on the Quality of English Language

Well written report

Author Response

Dear Reviewer,

Thank you for your comments and suggestions.

The citated paper, refers to “patients with advanced ovarian cancer who had undergone intraabdominal macroscopically complete resection and had normal lymph nodes both before and during surgery”. In our series, we did not talk about systematic dissection of normal lymph nodes, but we consider patients with bulky metastatic nodes. In primary surgery, the removal of bulky nodes does not have a staging purpose but enables to reduce macroscopic disease residue to zero. On the other hand, in case of recurrent disease this procedure is crucial for obtaining histological confirmation of the disease, having fundamental implications for further medical therapies, particularly PARP-inhibitors.

The extension of the disease was intraoperatively evaluated and reported according to the Fagotti score. We clarified this point in the text.

Surgical technique for laparoscopic removal of bulky para-aortic nodes without repositioning surgical field during laparoscopic debulking for advanced ovarian cancer. Puppo A, Olearo E, Ceccaroni M. Facts Views Vis Obgyn. 2022 Jun;14(2):189-191. doi: 10.52054/FVVO.14.2.029.

This paper was a case-report about laparoscopic removal of bulky para-aortic nodes without repositioning surgical field in advanced ovarian cancer. Therefore, the purpose of the present paper was different.

“Things Have Changed"-Laparoscopic Cytoreduction for Advanced and Recurrent Ovarian Cancer: The Experience of a Referral Center on 108 Patients. Ceccaroni M, Roviglione G, Bruni F, Dababou S, Venier M, Zorzi C, Salgarello M, Ruffo G, Alongi F, Gori S, Driul L, Uccella S, Barra F.Cancers (Basel). 2023 Dec 6;15(24):5726. doi: 10.3390/cancers15245726.

This was a multicenter study about the feasibility of laparoscopic cytoreduction in advanced and recurrent ovarian cancer. Our study aims to specifically investigate the minimally invasive approach in advanced and recurrent ovarian cancer with bulky nodes in two specialized Italian centers, one of them not included in the citated study.

Reviewer 2 Report

Comments and Suggestions for Authors

Interesting paper on the role of MIS in ovarian cancer surgery. Relatively small cohort, but regarding ovarian cancer tumor biology, it is difficult to find a subset of patients with no carcinomatosis, but positive lymph nodes,

Please specify if you have performed a peritoneal cytology and, if positive, the result would influence the surgical strategy.

In Results part, is a confusion between primary surgery and primary debulking surgery: 16 patients underwent primary surgery, and not PDS.

Comments on the Quality of English Language

One single mistake: levator ani muscle, not elevator

Author Response

Dear Reviewer,

Thank you for your comments.

In all cases peritoneal cytology was performed, but the analysis was not conducted intraoperatively and it did not change our approach. We added this part in the text.

We changed the terms “PDS” with “primary surgery” in the Results.

We changed the term “levator ani” with “levator ani”.

Reviewer 3 Report

Comments and Suggestions for Authors The authors investigated the principles of minimally invasive surgery for advanced and recurrent ovarian cancer with nodal involvement. They analyzed the clinical results in this situation. In 21 patients who underwent laparoscopic surgery for advanced or recurrent ovarian cancer with nodal metastases, the minimally invasive approach proved to be a safe and promising technique with low complication rates and favorable clinical outcomes.
The stregnth:
the article is well structured and organized, it represents a further step forward in the application of minimally invasive surgery in daily practice. The conclusions are clear, the authors are self-critical and are aware of the limitations of the study (small number of patients, short follow-up time, non-randomized patients). However, the article shows the advantages of minimally invasive surgery in well-selected cases, which can be of great benefit to a specific patient group. Weakness: poor images quality (resolution)

Author Response

Dear Reviewer,

Thank you for your feedback.

We improved the images quality, as suggested.

Reviewer 4 Report

Comments and Suggestions for Authors

This is a well-written manuscript that examines the efficacy and feasibility of laparoscopic treatment of bulky lymph nodes in women with ovarian cancer. Nevertheless, this study had several limitations.

1. lack of control group (abdominal surgery)

2. the definition of bulky lymph nodes was unclear (approximately half of cases may not be bulky)

3. short follow-up period

Please add these limitations to the Discussion section.

Comments on the Quality of English Language

The quality of English was acceptable.

Author Response

Dear Reviewer,

Thank you for your constructive comments and suggestions.

The purpose of the present study was not to make a comparison between open and minimally invasive surgery, but to present and analyze our experience. This is explained in the last paragraph of the Discussion.

We are aware of the short follow up period (25 months) and we underlined the need for further analysis with longer follow up to obtain more valuable survival outcomes.

We added these limitations to the discussion.

In gynecologic oncology, a specific definition of “bulky nodes” is lacking. For non-small cell lung cancer bulky nodes are defined as lymph nodes larger than 3 cm, but in ovarian cancer no dimensional cut off was defined. Thus, commonly lymph nodes are considered “bulky” when their size is more than 10 mm with radiological abnormal appearance or abnormal metabolism at PET-scan. We added this definition in the text.

Reviewer 5 Report

Comments and Suggestions for Authors

Congratulations to the authors for the article and for their surgical activity. Here are a few observations that I believe require a response from them:

1.Preoperative diagnostic elements are mandatory upon which the recommendation for surgical intervention was based.

2.Similarly mandatory are the conditions and the method of selecting patients for minimally invasive surgery.

3.Was the decision made in consensus with the oncologist, possibly in a tumor board?

4.What was the surgical indication, specifically what did the surgical intervention target? Was it an extensive lymph node dissection in each case, or was only the group of lymph nodes with metastatic appearance targeted? In other words, was the presented technique applied similarly to all patients? If lymph node excision was performed, what was the rate of lymph node invasion, the number of excised lymph nodes, etc.?

5.What was the time required for the surgical interventions, both minimum and maximum, as well as the average?

6.The article is not a review and does not review the literature. It is just a normal format for an original article with bibliographic references from the literature.

7.The pictures used have very poor quality, consequently being inconclusive. I suggest replacing them with high-resolution ones.

Author Response

Dear Reviewer,

Thank you for your interesting suggestions.

1-2-3. Imaging played a fundamental role in the selection of patients. In all cases of primary surgery, a preoperative CT-scan was performed to clearly understand the extension and localization of the disease, while patients with recurrent disease also received a PET-scan. Subsequently, each case was discussed in the tumor board, in which surgeons, oncologists, radiologists, radiotherapists and pathologists were present. In our experience, a diagnostic laparoscopy was first performed to accurately examine the abdomen and to evaluate the presence and the extension of peritoneal carcinomatosis. In case of resectable disease, but in presence of large pelvic masses, extended carcinomatosis, need for multiple bowel resections or anesthesiologic controindication to laparoscopy an open approach was preferred, whereas a previous laparotomy or the presence of diaphragmatic carcinomatosis did not represent contraindications to MIS. Regarding recurrent disease, the minimally invasive approach was preferred in case of single-site recurrence or in case of oligometastatic but surgically accessible disease.

  1. In our patients, we did not performed an extensive lymph nodal dissection for staging purpose, but a targeted removal of bulky nodes and adjacent locoregional nodes. We clarify this aspect in the text.
  2. The median lenght of surgery was 180 minutes (range 85-250 minutes).
  3. The present study was not intended to be a review. The aim was to present and discuss our experience according to the current literature. We clarified this aspect in the text and we changed the title of the paper.
  4. We improved the images quality, as required.

Round 2

Reviewer 5 Report

Comments and Suggestions for Authors

Thank you to the authors for the additions made, which are much appreciated.

I believe it is necessary to add information regarding the number of excised lymph nodes in patients with primary disease, especially since the intervention involved significant expansion - para-aortic, para-caval, and inter-aortic-caval dissection - as well as the rate of lymph node invasion (number of invaded nodes) identified in the pathological anatomy. Additionally, were the lymph nodes excised in lymph node groups? If so, which groups were most affected?

Author Response

Thank you for your comment.

We added the information required in the text.